# New insights in the allelopathic traits of different barley genotypes: Middle Eastern and Tibetan wild-relative accessions vs. cultivated modern barley

Mauro Maver[1]*, Begoña Miras-Moreno[2], Luigi Lucini[2], Marco Trevisan[2], Youry Pii[1], Stefano Cesco[1], Tanja Mimmo[1]

1 Faculty of Science and Technology, Free University of Bozen-Bolzano, Bolzano, Italy, 2 Department for Sustainable Food Process, Università Cattolica del Sacro Cuore, Piacenza, Italy

* mauro.maver@unibz.it

**Data Availability Statement:** All relevant data are within the manuscript and its Supporting Information files.

## Abstract

The two alkaloids gramine and hordenine have been known for playing a role in the allelopathic ability in barley (*Hordeum vulgare* L.). These allelochemicals can be both found in leaves and roots in some barley cultivars whereas in others one seems to exclude the other. In this study eighteen accessions of barley from the Middle-East area, one accession from Tibet and the modern spring cultivar Barke, already used as parental donor in a nested associated mapping (NAM) population, were screened for their gramine, hordenine and N-methyltyramine (the direct precursor of hordenine) content in leaves, roots and exudates. Moreover, the toxicity of the three allelochemicals on root growth inhibition on lettuce (*Lactuca sativa* L.) was evaluated. Results of this study showed the preferential production of gramine and hordenine in leaves and roots, respectively, in the nineteen barley accessions. On the other hand, in the modern barley cultivar Barke, the highest content of hordenine in roots and the general lack of gramine suggests a favored biosynthesis of the former. Gramine was not detected in the root exudates. In additions, different metabolomic profiles were observed in wild relatives compared to modern barley genotypes. The results also showed the phytotoxic effects of the three compounds on root growth of lettuce seedlings, with a reduction in root length and an increase of root surface area and diameter. In conclusion, this study highlighted the impact of the domestication effects on the production and distribution of the two allelopathic alkaloids gramine and hordenine in barley.

## Introduction

Weeds represent a serious and complex issue in agriculture due to their high ability to interfere with plant growth and development causing both yield loss and quality [1]. Yet, hand weeding and mechanical weeding are not always effective and might be labour intensive, expensive and time consuming; in this context the use of synthetic herbicides was rapidly adopted worldwide

**Funding:** The author(s) received no specific funding for this work.

**Competing interests:** The authors have declared that no competing interests exist.

[2]. Also the chemical herbicide application has some major constraints due to the massive use of chemicals increased the selection pressure on weed communities [3–5]. Furthermore, problems as herbicides' persistence in soil, environmental contamination and pollution are the main side-effects of this management practice [6,7]. Therefore, since the use of synthetic herbicides has severely affected the sustainability of the agricultural production systems and due to a lack of new, efficient, environmental and crop safe synthetic compounds [8,9], in the recent years non-synthetic chemical alternatives have been gained increased interest [1,10].

For the abovementioned reasons, allelopathy, the ability to influence positively or negatively the surrounding area through the release of allelochemicals, gained centre stage as an attractive tool to naturally control weeds. Crops that have an allelopathic potential could be strategically adopted in weed management under field conditions in different ways, from the direct (intercropping and cover cropping systems) and indirect (dead material) release of allelochemicals [11,12], to the utilization of the latter as bioherbicides [13]. The allelopathic ability has been well described and characterized only in the last century and firstly defined in 1937 by Hans Molish [14] even though it has been gradually reconsidered and integrated until the present days. The up-to-date definition includes negative and positive influence on growth and development exerted by plants, micro-organisms, fungi and virus by secreting allelochemicals into the environment, *i.e.* chemical compounds belonging mainly to the secondary metabolism [15,16]. There are several important crops, e.g. *Avena sativa* L. (oat), *Triticum aestivum* L. (wheat), *Hordeum vulgare* L. (barley), *Oryza sativa* L. (rice) and *Zea mays* L. (corn), that have been known for centuries for their particular trait in suppressing weeds [17]. Even though exploiting plants with allelopathic traits might contribute to a more environmentally sustainable weed management, transfer this useful ability in high yield agronomical plants has to be evaluated very carefully, since an improved allelopathic effect, as intended as increased production of secondary metabolites, might limit high yield outputs [12,18,19]. The ability of barley to suppress and contrast weed growth has been known for centuries but so far, little is known about its main allelochemicals which have been demonstrated having phytotoxic effects [20]. Among them, two alkaloids, gramine (N,N-dimethyl indole methylamine) and hordenine (N,N-dimethyltyramine) were the first compounds proposed to account for the main role in the allelopathic ability of barley [21,22]. Besides negative effects against weeds and other plants, there is evidence that gramine and hordenine have also roles in defence in response to abiotic and biotic stresses. For instance, gramine accumulation has shown to be induced in barley leaves due to an increase of growth temperature [23], drought [24], attacks of aphids [25–28] and fungi [29]. Yet, despite several reports based on the induction of gramine production, the pathway of signal transmission is still largely unknown [29]. On the other hand, hordenine was demonstrated to trigger the plant defence response through the jasmonate-dependent defence pathway [30]. Lovett and Hoult (1995) already observed that the domestication process and the breeding selection necessary to maintain best agronomic traits within barley germplasm might have reduced the gramine synthesis in favour of hordenine [31]. Consequently the allelopathic potential of barley against weeds [32,33] is strongly reduced compared to wild barley relatives [34]. In some cases the ability of accumulating gramine was even lost or extremely reduced as demonstrated in Proctor, Morex and Barke barley cultivars [35,36]. Conversely, wild barley and Middle Eastern landraces that spontaneously grow in that area are considered the primary center of barley domestication [37,38] and have proven to be a rich source of genetic variability [39,40], can accumulate gramine. Regarding this, with the increased availability of genetic and genomic resources for barley [41,42] it is now possible to investigate the molecular basis of gramine and other secondary metabolites at an unprecedented depth. For instance, by tapping into experimental populations between domesticated and wild barley genotypes, such as NAM population HEB-25 analyzing the parental donors

selected for the development of the NAM population HEB-25 [43,44], novel insights into the inheritance of allelopathy in barley can revealed and new strategies toward sustainable weed management developed. On the basis of these premises the present research aims at i) assessing the gramine, hordenine and N-methyltyramine (the last precursor in hordenine pathway) content in the root and leaf tissues of 20 wild barley accessions from Middle-East area and the spring barley elite cultivar Barke, ii) investigating the growth dependent production and exudation profile of the three compounds, and iii) evaluating the phytotoxic effect of gramine, hordenine and N-methyltyramine on *Lactuca sativa* L.

## Material and methods

### Plant materials

Seeds of spring barley elite cultivar Barke (*Hordeum vulgare* ssp. *vulgare*, hereafter *Hv*) and 20 barley accessions (Hordeum identity, HIDs), which comprise 19 wild barley accessions of *H. vulgare* ssp. *Spontaneum* (*Hsp*), the progenitor of domesticated barley and one Tibetian *H. vulgare* ssp. *Agriocrithon* (*Hag*) accession were selected [43]. Wild accessions were chosen as representatives of the high genetic diversity that still occurs in the region of the Middle East, one of the main centres of barley domestication (S1 Table).

### Growth conditions and sampling

Barley seeds were germinated in the dark on a filter paper moistened with 0.5 mM $CaSO_4$. After 4 days, homogeneous seedlings were transferred to a complete nutrient solution (Ca$(NO_3)_2$ x $4H_2O$ 2Mm; $MgSO_4$ x $7H_2O$ 0.5Mm; $K_2SO_4$ 0.7Mm; KCl 0.1Mm; $KH_2PO_4$ 0.1mM; $H_3BO_3$ 1µM, $MnSO_4$ x $H_2O$ 0.5µM; $CuSO_4$ 0.2µM; $ZnSO_4$ x $7H_2O$ 0.5µM; $(NH_4)_6Mo_7O_{24}$ x $4H_2O$ 0.01µM; Fe-EDTA 100µM) and grown hydroponically under continuous aeration for other 4 days under controlled conditions in a climate chamber with a 14/10 h day/night regime, 24˚C/19˚C, 70% relative humidity and 250 µmol $m^{-2}$ $s^{-1}$ light intensity as described by [45]. The first leaf and whole root system roots were sampled at 2 and 4 days separately and extracted in 1 mL of pure methanol for 24 hours under continuous shaking as previously described by [25].

### Chromatographic determination of gramine, hordenine and N-methyltyramine

Plant tissue extracts were analyzed for their gramine, hordenine and N-methyltyramine content by a modified method [46], using a Waters ALLIANCE HPLC system with autosampler coupled to a Waters PDA Detector and a Waters ACQUITY QDA, equipped with an electrospray ionization (ESI) interface in positive ionization (PI) mode. A LiChrospher RP-18 column 250 mm x 4.0 mm, 5µm (Phenomenex, USA) was used as column and a gradient elution with a flow rate of 0.6 mL $min^{-1}$ was applied. Two mobile phases, A and B, were used for the gradient elution: mobile phase A consisted of acetonitrile, 0.1% formic acid (Sigma-Aldrich, >99%) while mobile phase B consisted of 0.01 M ammonium acetate 0.1%, formic acid (Sigma-Aldrich, >99%) in water. The gradient started with 10% A and 90% of B, increased to 50% A within 10 minutes and up to 90% A at 25 min, then returned to 10% A in 5 minutes, plus other 10 minutes of gradient stabilization. The injected volume was 20 µL and the UV-absorption was monitored at 275 nm with a resolution of 1.2 nm. The optimized parameters of the QDA interface were: source temperature, 120˚C; vaporizer temperature, 600˚C; drying gas (nitrogen) temperature, 600˚C; cone voltage, 15V; capillary voltage, 800 V; Gain 1. For the data collection the SIR modus was adopted by monitoring the m/z 175 and 130 for gramine as

common main fragment, m/z 166 for hordenine and 121 for N-methyltyramine. The retention times were 12.1 min, 7.7 min and 6.7 min, for gramine, hordenine and N-methyltyramine, respectively. All the three standards were purchased from Sigma-Aldrich (>99%) and stock solution prepared in methanol were used for the quantifications. Limit of detection (LOD) for gramine, hordenine and N-methyltyramine were 1.42 μM (0.24 mg $L^{-1}$), 0.23 μM (0.038 mg $L^{-1}$) and 0.21 μM (0.033 mg $L^{-1}$), respectively.

## Determination of growth dependent metabolites biosynthesis and exudation

Four days old barley seedlings of Barke, HID-102 and HID-219 were transferred to 50 mL pots containing a complete nutrient solution and grown for another 3 days. During these 3 days, root exudates were collected every 6 hours. Plants were thereby transferred to 15 mL falcon tubes (one plant per falcon tube) containing 10 mL milliq-water for 4 hours. The trap solutions were continuously aerated. After 4 hours, the solution containing the exudates was filtered, frozen at -20˚C, lyophilized and finally resuspended in 0.75 mL of pure methanol. Prior HPLC analysis samples were again filtered with a 0.45 μm filter (0.45μm Syringe Filters, Phenomenex). Leaves and roots of each plant were collected as well and processed as described above.

## Untargeted metabolomics

The screening of metabolites of root and shoot extracts of Barke, Solist, HID-380, HID-055 and HID-219 was performed through ultra-high-pressure liquid chromatographic system (Agilent 1200 series) coupled to a high-resolution mass spectrometry (quadrupole-time-of-flight mass spectrometry, Agilent 6550 iFunnel) system equipped with a JetStream electrospray source (UHPLC-ESI/QTOF-MS). Acquisition conditions were optimized in previous experiments [47]. Briefly, an Agilent Zorbax Eclipse-plus C18 column (100 × 2.1 mm, 1.8 μm) was used for reverse-phase chromatography utilizing a binary gradient of methanol and water for 33 min, with a flow of 200 μL $min^{-1}$ at 35˚C. The mass spectrometric acquisition was done in SCAN (100–1,000 m/z) and positive polarity. Raw data were processed by the software MassHunter Profinder B.06 (from Agilent Technologies) using the "find-by-formula" algorithm. Features annotation was then based on monoisotopic accurate mass and isotope pattern (exact masses with accuracy < 5 ppm, relative abundances and m/z spacing) and expressed as overall identification score. Compounds were putatively annotated using both monoisotopic accurate mass and isotopes pattern (accurate spacing and isotopes ratio), using the software Profinder B.07 (Agilent technologies) and the database PlantCyc 12.6 (Plant Metabolic Network, http://www.plantcyc.org; downloaded April 2018). Thereafter, the raw data were re-processed against an *in-house* database, manually curated to include the intermediate biosynthetic compounds and the end products for gramine, hordenine and N-methyltyramine. Elaboration of metabolomics dare were formerly carried out using Agilent Mass Profiler Professional B.12.06 (from Agilent Technologies) as described by [48]. Compounds were filtered by abundance (area >10000 counts), normalization at the 75th percentile and baselined to the median of the control. Unsupervised hierarchical cluster analysis was carried out from the fold-change based heatmaps, setting the similarity measure as 'Euclidean' and using 'Wards' as the linkage rule. Thereafter, the dataset was exported into SIMCA 13 (Umetrics, Malmo, Sweden), UV-scaled and elaborated for Orthogonal Projections to Latent Structures Discriminant Analysis (OPLS-DA) supervised modelling. The model parameters (goodness-of-fit R2Y and goodness-of-prediction Q2Y) were calculated and the model validated through cross-validation CV-ANOVA ($p < 0.01$), whereas overfitting was excluded by permutation testing (n = 100). Outliers were finally investigated using Hotelling's T2 (95% and 99% confidence limits for suspect and

strong outliers, respectively). Variables of Importance in Projection (VIP analysis) was finally used to identify discriminating compounds (VIP score > 1.6).

## Phytotoxicity assays

The phytotoxicity potential of the three metabolites, gramine, hordenine and N-methyltyramine was investigated by measuring the main growth root parameters of lettuce (*Lactuca sativa* L.) once treated with the three alkaloids. Seeds of lettuce were placed in petri dishes laid out with filter paper (Whatman N˚41, Whatman, Maidstone, UK) and soaked with 1 mL of solution of each alkaloid. The solutions were previously prepared from 5 mM MES buffer in distilled water adjusted to pH 6.15 by adding NaOH as described in [49], in which gramine, hordenine and N-methyltyramine were added in order to reach concentrations of 0.5 mM and 1 mM, obtaining different treatments. Controls were also prepared using the buffer alone. Petri dishes were sealed with Parafilm and incubated in the dark in the climate chamber for 48 hours. Thereafter, the germinated seeds were counted, and the main growth parameters were measured to calculate the inhibition percentage of the three metabolites investigated on *Lactuca sativa* L.. Root parameters were assessed by scanning the seedlings with WinRHIZO™ system (WinRhizo software, EPSON1680, WinRHIZO Pro2003b, Regent Instruments Inc., Quebec, Canada).

## Statistical analysis and visualization

The results are reported as mean ± SE. The significance of differences among genotypes/times/treatments means was calculated by One-way ANOVA with post-hoc Tukey HSD with α = 0.05 using R software (version 3.6.0). The following R packages were used: for data visualization ggplot2 v.3.2.0 [50] and Agricolae v.1.3–1 for Tukey post-hoc test [51].

# Results

## UHPLC-QTOF-MS discrimination of wild-relative accessions vs cultivated modern barley

First of all, the metabolic signatures of the five barley genotypes were investigated using an untargeted metabolomics approach based on UHPLC-ESI/QTOF-MS profiling, in order to investigate the differences between the wild-relative accessions and cultivated modern barley. Overall, this analytical approach allowed annotating more than 2500 compounds, considering both leaves and roots.

Multivariate statistical techniques have been applied to differentiate barley cultivars at molecular level. A preliminary hierarchical cluster analysis, based on the fold change of metabolites, was produced as unsupervised approach (S1 Fig). As expected, the results showed that the samples clustered by organ (leaves or roots) more than cultivars. Therefore, to better understand differences in barley profiling, roots and leaves were analyzed separately.

A supervised OPLS-DA multivariate statistical approaches allowed to perfectly separate the wild accessions from the cultivated modern barley, in both leaves and roots (Fig 1A and 1B). The model cross-validation parameters were considered as acceptable. In particular, goodness-of-fit R2Y was 1 in both cases and prediction ability Q2Y was 0.906 for leaves and 0.881 for roots. The cross-validation ANOVA resulted in a Fischer's probability < 0.001 for both OPLS-DA models, no outlier samples could be observed by Hotelling's T2 and permutation test excluded overfitting. Afterwards, the variables importance in projection of the OPLS-DA model (VIP analysis), which calculate how a variable contributes to the model, were identified considering VIP scores of > 1.6. VIP compounds are included in S2 Table for leaves and S3

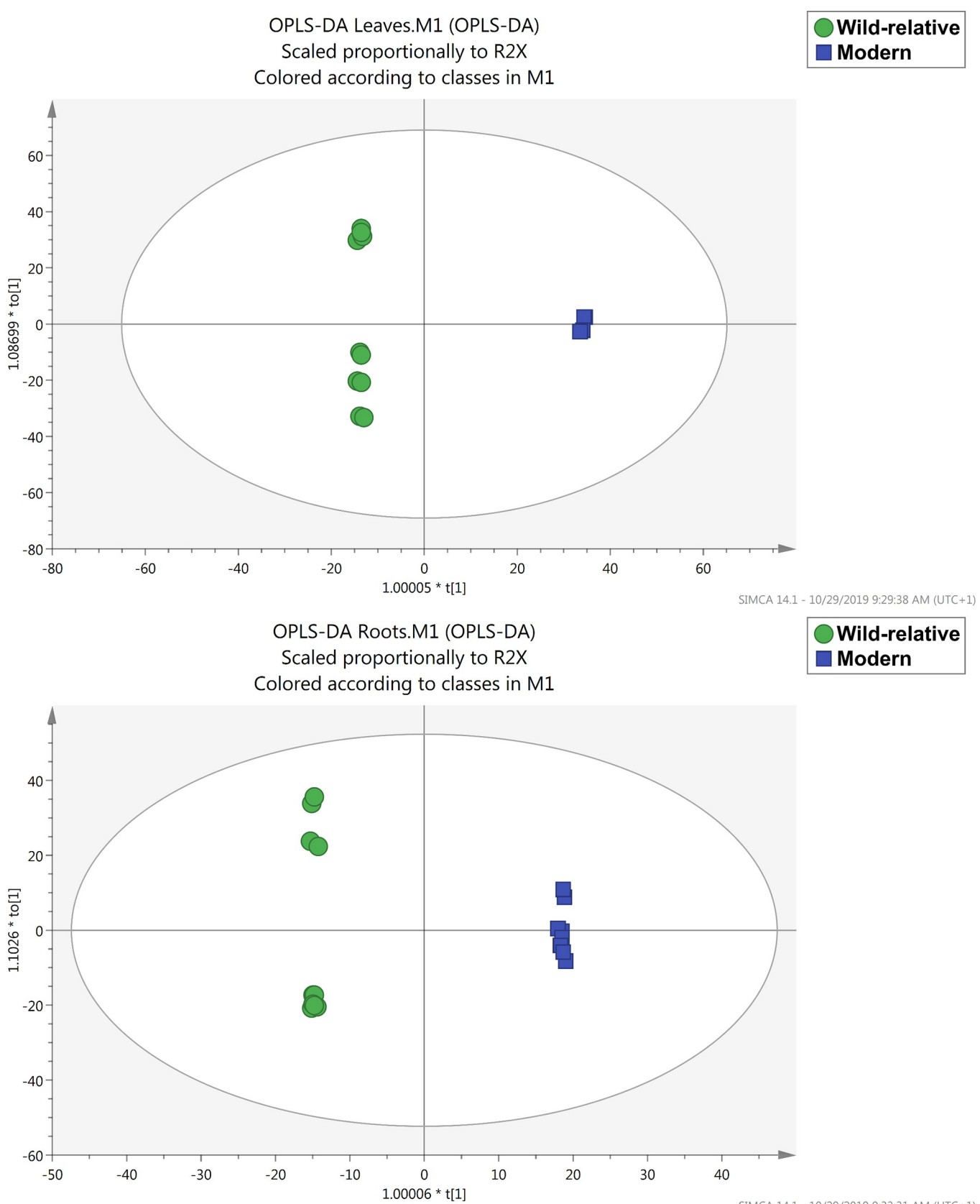

**Fig 1. Supervised OPLS-DA multivariate statistical.** Orthogonal Projections to Latent Structures Discriminant Analysis (OPLS-DA) on barley leaves (A) and roots (B) phenolic profile according to their origin.

Table for roots. VIP markers were classified into biochemical classes according to the PlantCyc database, in order to highlight the classes providing the major contribution to discrimination in the OPLS-DA supervised model. VIP markers showed secondary metabolites as discriminant compounds, in the metabolic profiles of leaves and roots from wild-relative accessions and cultivated modern barley. In fact, alkaloids, phenolic compounds and phytohormones were the most represented classes responsible for the separation in the OPLS-DA model.

## Gramine content in barley tissues

Gramine has been detected in all 20 wild barley accessions, 19 belonging to *H. vulgare* ssp. *Spontaneum* and 1 to *H. vulgare* ssp. *Agriocrithon*, both in the first leave and in roots (Table 1). Gramine could not be detected in the plant tissues of the spring barley elite cultivar Barke (<LOD). In general, gramine concentration resulted about 50-fold higher in leaves compared to roots. In leaves, the In group showed the highest gramine content in both the sampling days, 5.290 µmol g$^{-1}$ FW after 2 days and 3.687 µmol g$^{-1}$ FW after 4 days respectively, whereas *H.ag* barley (Tibet) shown the significative lowest content both at 2 and 4 days, 2.548 µmol g$^{-1}$ FW and 2.032 µmol g$^{-1}$ FW. In roots, Iq-In and T-G groups showed the highest gramine content at 2 and 4 days (0.066 µmol g$^{-1}$ FW and 0.095 µmol g$^{-1}$ FW), whereas *H.ag* barley (Tibet) the lowest content at both days, 0.011 µmol g$^{-1}$ FW and 0.018 µmol g$^{-1}$ FW respectively. Comparing the gramine content at 2 and 4 days, in leaves has been observed a general decreasing (approx. -20%) whereas in roots the trend is in some cases the opposite. Leaves resulted as the main plant tissue with the highest content of gramine in barley.

## N-methyltyramine content in barley tissues

The direct precursor of hordenine, N-methyltyramine, has been detected in all the barley plants analyzed (Table 2). In leaves, the highest content was observed in the In group both at 2 and 4 days with 0.024 µmol g$^{-1}$ FW and 0.027 µmol g$^{-1}$ FW respectively, whereas the lowest

**Table 1. Gramine content measured in the leaf and root extracts of 2 and 4 days old barley plants.**

| | Gramine | | | |
|---|---|---|---|---|
| | µmol g$^{-1}$ (±SE) FW | | | |
| | Leaf | | Root | |
| Origin group | 2d | 4d | 2d | 4d |
| Barke | <LOD | <LOD | 0.005[b] (±0.002) | 0.015[b] (±0.005) |
| H.ag (Tibet) | 2.548[c] (±0.191) | 2.032[c] (±0.305) | 0.011[b] (±0.006) | 0.018[b] (±0.004) |
| I-J | 3.482[bc] (±0.225) | 2.745[bc] (±0.217) | 0.049[ab] (±0.011) | 0.032[b] (±0.004) |
| In | 5.290[a] (±0.262) | 3.687[a] (±0.223) | 0.030[ab] (±0.008) | 0.028[b] (±0.011) |
| Iq-In | 3.791[bc] (±0.180) | 3.121[abc] (±0.170) | 0.066[a] (±0.007) | 0.056[ab] (±0.004) |
| L-WS | 4.329[bc] (±0.225) | 3.257[ab] (±0.183) | 0.065[a] (±0.008) | 0.054[b] (±0.007) |
| T-G | 4.033[b] (±0.218) | 3.372[ab] (±0.143) | 0.045[ab] (±0.006) | 0.095[a] (±0.020) |
| T-NS | 3.545[bc] (±0.196) | 2.621[bc] (±0.134) | 0.040[ab] (±0.006) | 0.056[ab] (±0.006) |

Gramine content measured in the leaf and root extracts of 2 and 4 days old barley plants (spring modern Barke and wild-relative accessions barley divided in origin groups). The specific origin of the barley accessions are listed in S1 Table. Data are expressed as mean ± SE, n = 10. Letters following the means indicate significant differences, One-way ANOVA with post-hoc Tukey HSD with α = 0.05.

**Table 2. N-methyltyramine content measured in the leaf and root extracts of 2 and 4 days old barley plants.**

| | N-methyltyramine | | | |
| | μmol g$^{-1}$ (±SE) FW | | | |
| | Leaf | | Root | |
| Origin group | 2d | 4d | 2d | 4d |
|---|---|---|---|---|
| Barke | 0.010$^b$ (±0.002) | 0.003$^c$ (±0.001) | 0.224$^d$ (±0.032) | 0.191$^d$ (±0.010) |
| H.ag (Tibet) | 0.009$^b$ (±0.004) | 0.025$^{ab}$ (±0.011) | 0.815$^{bc}$ (±0.022) | 0.687$^{bc}$ (±0.218) |
| I-J | 0.016$^{ab}$ (±0.002) | 0.008$^c$ (±0.001) | 0.689$^{cd}$ (±0.086) | 0.581$^{bcd}$ (±0.086) |
| In | 0.024$^a$ (±0.004) | 0.027$^a$ (±0.002) | 1.389$^a$ (±0.102) | 1.137$^a$ (±0.099) |
| Iq-In | 0.018$^{ab}$ (±0.002) | 0.009$^c$ (±0.002) | 1.069$^{ab}$ (±0.124) | 0.713$^b$ (±0.064) |
| L-WS | 0.019$^{ab}$ (±0.001) | 0.009$^c$ (±0.001) | 0.604$^{cd}$ (±0.043) | 0.453$^{bcd}$ (±0.031) |
| T-G | 0.014$^{ab}$ (±0.001) | 0.012$^{bc}$ (±0.003) | 0.485$^{cd}$ (±0.059) | 0.506$^{bcd}$ (±0.064) |
| T-NS | 0.018$^{ab}$ (±0.001) | 0.008$^c$ (±0.001) | 0.535$^{cd}$ (±0.030) | 0.392$^{bcd}$ (±0.035) |

N-methyltyramine content measured in the leaf and root extracts of 2 and 4 days old barley plants (spring modern Barke and wild-relative accessions barley divided in origin groups). The specific origin of the barley accessions are listed in in S1 Table. Data are expressed as mean ± SE, n = 10. Letters following the means indicate significant differences, One-way ANOVA with post-hoc Tukey HSD with α = 0.05.

content was detected in *H.ag* barley (Tibet) at 2 days (0.009 μmol g$^{-1}$ FW) and in Barke at 4 days (0.003 μmol g$^{-1}$ FW). In general, a not uniform decrease was observed in leaves (from -15% to -70%) a part of an increase in *H.ag* barley (Tibet). In roots, the highest content was detected in the In group in both the sampling days with 0.689 μmol g$^{-1}$ FW and 1.137 μmol g$^{-1}$ FW respectively, whereas the lowest content of N-methyltyramine was detected in the modern cv Barke, (0.224 μmol g$^{-1}$ FW and 0.191 μmol g$^{-1}$ FW at 2 and 4 days, respectively). In this case an increase was observed only in T-G group. In general, N-methyltyramine content has been observed being about 50-fold higher in roots compared to leaves, resulting as the main plant tissue with the highest content of this metabolite.

## Hordenine content in barley tissues

Hordenine has been detected in all the barley accessions and in Barke (Table 3) but could not be detected in Barke leaves (<LOD). In leaves, the In group revealed the highest content at 2

**Table 3. Hordenine content measured in the leaf and root extracts of 2 and 4 days old barley plants.**

| | Hordenine | | | |
| | μmol g$^{-1}$ (±SE) FW | | | |
| | Leaf | | Root | |
| Origin group | 2d | 4d | 2d | 4d |
|---|---|---|---|---|
| Barke | <LOD | <LOD | 0.409$^a$ (±0.039) | 0.225$^a$ (±0.021) |
| H.ag (Tibet) | 0.018$^b$ (±0.001) | 0.006$^c$ (±0.002) | 0.179$^{abc}$ (±0.032) | 0.177$^{ab}$ (±0.125) |
| I-J | 0.017$^b$ (±0.002) | 0.010$^{bc}$ (±0.001) | 0.020$^c$ (±0.005) | 0.019$^b$ (±0.003) |
| In | 0.029$^a$ (±0.004) | 0.018$^b$ (±0.004) | 0.232$^{ab}$ (±0.047) | 0.148$^{ab}$ (±0.026) |
| Iq-In | 0.014$^b$ (±0.002) | 0.006$^c$ (±0.001) | 0.157$^{bc}$ (±0.032) | 0.107$^{ab}$ (±0.030) |
| L-WS | 0.018$^b$ (±0.002) | 0.011$^{bc}$ (±0.002) | 0.232$^{ab}$ (±0.027) | 0.090$^b$ (±0.014) |
| T-G | 0.010$^b$ (±0.002) | 0.012$^{bc}$ (±0.002) | 0.239$^{ab}$ (±0.020) | 0.125$^{ab}$ (±0.009) |
| T-NS | 0.015$^b$ (±0.002) | 0.007$^c$ (±0.001) | 0.241$^{ab}$ (±0.041) | 0.112$^{ab}$ (±0.019) |

Hordenine content measured in the leaf and root extracts of 2 and 4 days old barley plants (spring modern Barke and wild-relative accessions barley divided in origin groups). The specific origin of the barley accessions are listed in in S1 Table. Data are expressed as mean ± SE, n = 10. Letters following the means indicate significant differences, One-way ANOVA with post-hoc Tukey HSD with α = 0.05.

and 4 days (0.029 µmol g$^{-1}$ FW and 0.018 µmol g$^{-1}$ FW respectively) whereas the lowest content of hordenine was determined in T-G group at 2 days (0.010 µmol g$^{-1}$ FW) and in *H.ag* barley (Tibet) and Iq-In group at 4 days (0.006 µmol g$^{-1}$ FW) (Table 3). In roots, the highest content of hordenine was observed in Barke for both the sampling days (0.409 µmol g$^{-1}$ FW and 0.225 µmol g$^{-1}$ FW at 2 and 4 days, respectively). In both the tissues a general but not uni-formed decrease was observed (from -30% to -60%) a part of the T-G group in leaves. The root-to-leaf ratio of hordenine content is around 10, confirming roots as the main plant tissue for the highest content of this metabolite.

## Growth dependent metabolites biosynthesis and release

A detailed characterization of the production of the three metabolites, gramine, hordenine and N-methyltyramine has been carried out in leaves, root and exudates within 3 days after germi-nation in Barke, HID-102 and HID-219. This analysis confirmed and mainly showed that hor-denine and its precursor N-methyltyramine (NMT) were released from the roots of all the barley lines, having a significant decrease starting already after six hours (t1). Barke also showed the highest content of hordenine in roots than NMT along all the samplings, whereas the opposite trend was observed in the wild barley accessions. In leaves, the content of both the compounds was fairly significant constant in mostly of the samplings, but in Barke leaves, hor-denine could not be detected (Fig 2).

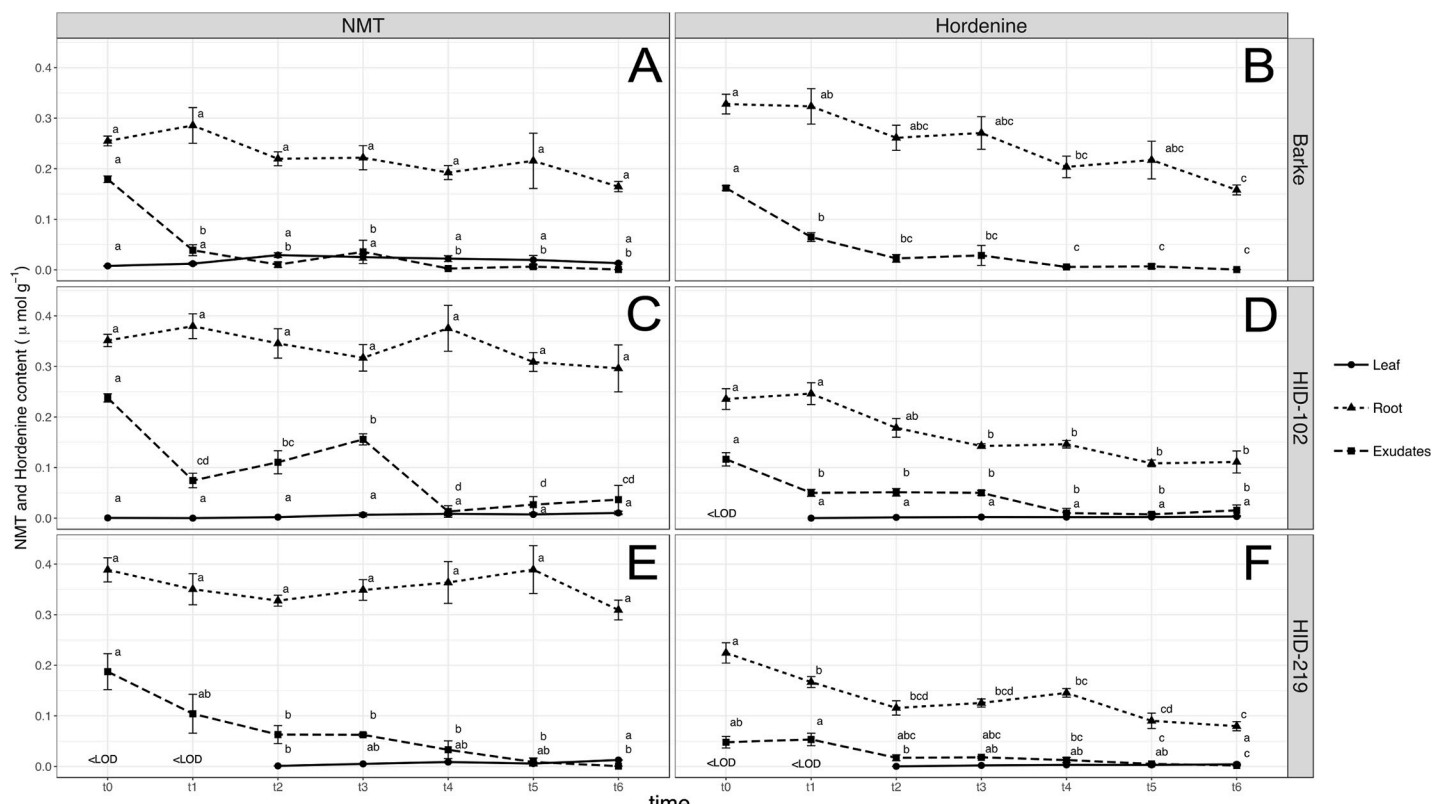

**Fig 2. Growth dependent N-methyltyramine (NMT) and hordenine biosynthesis and release.** N-methyltyramine (NMT) and hordenine content monitored in indicated tissues or exudates every 6 hours after 4 days of germination of (A, B) spring modern Barke and (C, D) wild-relative accessions barley HID-102 and (E, F) HID-219. Data are expressed as mean ± SE, n = 5, where t0 = 4 days after germination and t1 = t0+6 hours. Letters following the means indicate significant differences, One-way ANOVA with post-hoc Tukey HSD with α = 0.05.

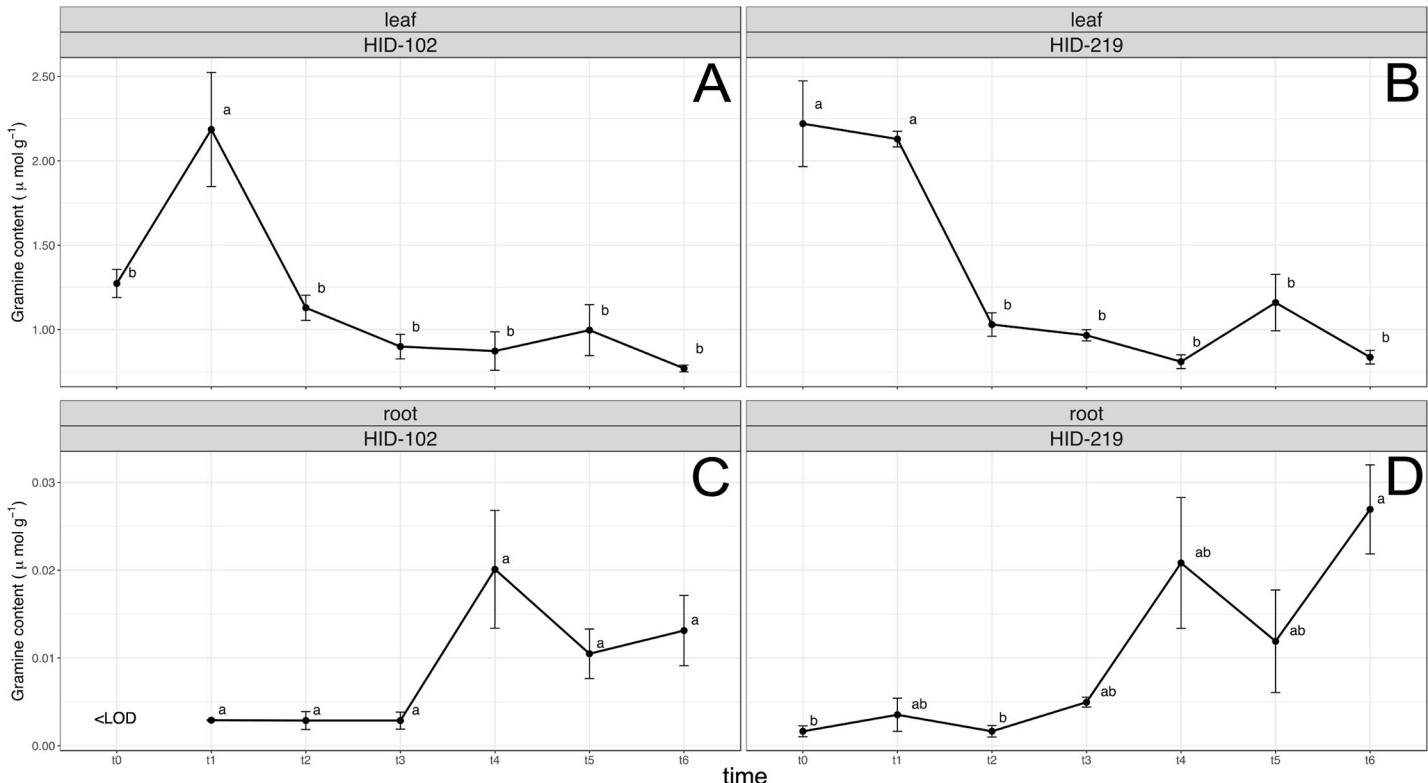

**Fig 3. Growth dependent gramine biosynthesis and release.** Gramine content monitored in (A, B) leaf and (C, D) root, respectively, every 6 hours after germination of wild-relative accessions barley HID-102 (L-WS) and HID-219 (Asia). Data are expressed as mean ± SE, n = 5. Letters following the means indicate significant differences, One-way ANOVA with post-hoc Tukey HSD with α = 0.05.

Gramine could not be detected in exudates but only in leaves and roots of wild barley accessions (Barke <LOD) (Fig 3). In particular, a very similar trend of hordenine production, in both the tissues, has been observed, with a decrease in leaf but an increase in root (only significant in HID-219) (Fig 3).

## Metabolomic targeted analysis

The metabolomic targeted analysis of roots were performed in barley roots of Barke, HID-055, HID-219 and HID-380 (Fig 4), focusing on the main and known compounds involved in the gramine biosynthesis pathway, *i.e.* regarding the main precursors of gramine, the amino acid tryptophan, no significant differences were observed among all the lines analyzed (Fig 4A). MAMI (N-methyl-aminomethylindole), the direct precursors of gramine, was the highest in Barke (Fig 4B) whereas gramine was the highest in HID-055, while it could be not detected in Barke (<LOD); HID-219 and HID-380 showed intermediate gramine concentration, around 45% lower than HID-055 (Fig 4C).

## Phytoxicity essays

The phytoxic effects of gramine, hordenine and N-methyltyramine were assessing main root parameters of lettuce (*Lactuca sativa* L.) in the presence of the three compounds at different concentrations (Table 4). After 48h, it can be seen that, in terms of mean root length, 1 mM hordenine has led to the highest inhibition percentage (-25.03%), followed by 1mM Gramine (-14.90%) and 1 mM N-methyltyramine (-6.39%); only in the presence of gramine a negative

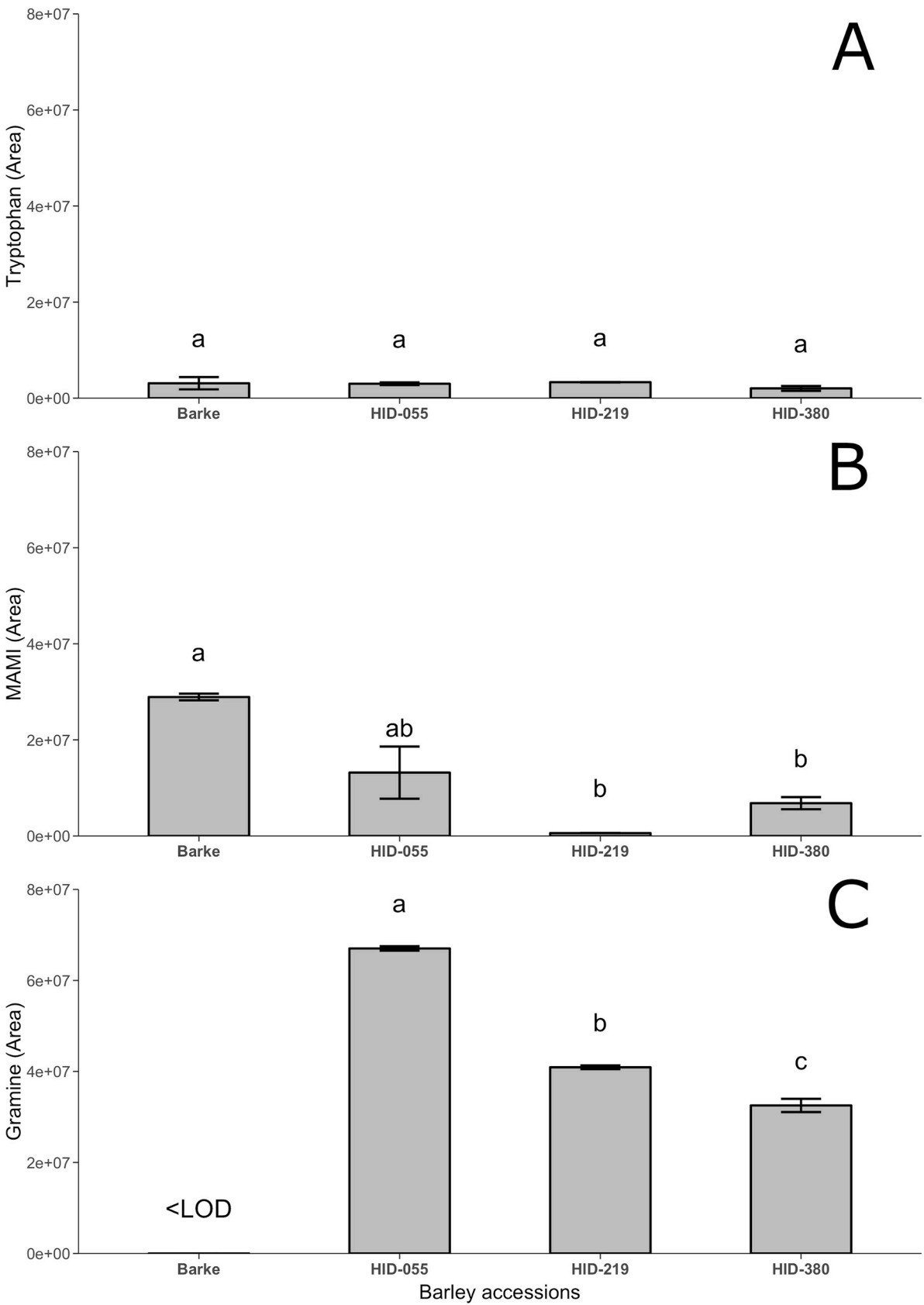

**Fig 4. Metabolomic targeted analysis.** Metabolomic targeted analysis of (A) Tryptophan, (B) N-methyl-aminomethylindole (MAMI) and (C) Gramine in barley roots 6 days after germination. Data are expressed as mean area ± SE, n = 3. Letters following the means indicate significant differences, One-way ANOVA with post-hoc Tukey HSD with α = 0.05.

inhibition percentage both at 0.5 and 1 mM was observed. Concerning the mean root surface and diameter, a general increase (approximately 20–25%) with all the treatments applied, compared to the control, was observed.

## Discussion

The two allelochemicals gramine and hordenine are the main alkaloids identified in barley with previously demonstrated allelopathic potential [20]. Nevertheless, only the hordenine pathway has been entirely characterized, having as a main precursor the amino acid tyrosine [52]. On the other hand, for gramine only tryptophan as the first amino acid and the last methylation steps are known within the biosynthetic pathway [36]. At the same time, a significant decrease or even a complete loss in gramine production in modern and cultivated barley if compared to wild-relatives barley has been demonstrated, most likely due to the domestication process [18,35,36,53]. Such a decrease of the biosynthesis has however not been observed for hordenine. Therefore, analyzing wild-relatives and barley landraces, which are still present and grown in the domestication area in the Middle-East [54] might provide new insights about these aspects. Indeed, in this study, wild barley accessions with the elite cv Barke, as parental donors involved in the development of the NAM population "Halle Exotic Barley" (HEB-25) [44], have been characterized for gramine, hordenine and N-methyltyramine biosynthesis. N-methyltyramine, although it has been isolated from barley in 1950 [55], only recently, taking advantage of new methodologies and lower limit of detections, *i.e.* by HPLC-ESI-MS/MS, it has been quantified [56]. Allelochemicals can be found in different parts within the plants while their production and release can also be induced by abiotic and biotic stress [16]. Depending on the kind of interactions the plant has to establish with the environment, allelochemicals are mainly produced and/or released from specific part of plants, *i.e.* leaves and stems for plant-insects/pests and roots for plant-rhizosphere, plant-plant interactions. Indeed root exudates, in addition to being important in plant interactions, can be released actively by roots as response as defense mechanisms, and through several ways, *i.e.* diffusion, vesicle transport and ion channels [57], although for allelochemicals this process has

**Table 4. Phytotoxicity effects of gramine, hordenine and N-methyltyramine on lettuce root.**

| Treatment | Mean root length (±SE) (cm) | Inhibition percentage | Mean root surface area (±SE) (cm²) | Inhibition percentage | Mean root diameter (±SE) (mm) | Inhibition percentage |
|---|---|---|---|---|---|---|
| Control | 1.21[a] (±0.03) | - | 0.20[d] (±0.01) | - | 0.54[b] (±0.01) | - |
| Gramine 0.5 mM | 1.15[ab] (±0.04) | -5.38% | 0.23[bc] (±0.01) | +15.52% | 0.66[a] (±0.01) | +21.88% |
| Hordenine 0.5 mM | 1.21[a] (±0.04) | +0.10% | 0.24[ab] (±0.01) | +19.65% | 0.64[a] (±0.01) | +18.98% |
| N-methyltyramine 0.5 mM | 1.28[a] (±0.04) | +5.86% | 0.27[a] (±0.01) | +33.43% | 0.68[a] (±0.01) | +26.33% |
| Gramine 1 mM | 1.03[bc] (±0.04) | -14.90% | 0.21[cd] (±0.01) | +1.21% | 0.64[a] (±0.01) | +17.39% |
| Hordenine 1 mM | 0.91[c] (±0.03) | -25.03% | 0.20[d] (±0.01) | -1.38% | 0.70[a] (±0.01) | +28.74% |
| N-methyltyramine 1 mM | 1.14[ab] (±0.05) | -6.39% | 0.23[bc] (±0.01) | +15.11% | 0.67[a] (±0.01) | +24.50% |

Phytotoxicity effects of gramine, hordenine and N-methyltyramine on mean root length, mean root surface area and mean root diameter. Data are expressed as mean ± SE. Letters following the means indicate significant differences, One-way ANOVA with post-hoc Tukey HSD with α = 0.05

not been well characterized yet if compared to the above-ground interactions [58]. The metabolomic analysis carried out in either roots and leaves clearly evidenced distinct signatures between wild relative accessions and modern barley. Although it can be postulated that these differences go far beyond the mere allelopathic activity, it is clear that secondary metabolism is responsible for such discrimination. In particular, alkaloids (together with phenolics) were among the most represented discriminating compounds. Target analysis evidenced that, among the three alkaloids evaluated, hordenine and its direct precursor N-methyltyramine were detected in the root exudates of wild relative accessions and modern Barke barley even if their concentrations decreased significantly with time (Fig 2). Gramine on the other hand could not be detected (Fig 2). It is interesting to note that while gramine was not detected in the root exudates, this alkaloid was measured in root extracts (Fig 3C and 3D). In this respect it should be highlighted that alkaloids like gramine and hordenine cannot simply diffuse through a phospholipid bilayer like the plasma membrane of healthy root cells [49]. However, it has been reported that the root release of gramine in the rhizosphere could be guaranteed by an altered cell membrane permeability like that induced by phenolic compounds or the activation of a specific transmembrane transport mechanism triggered by a particular stress [57,58]. A decomposition process of plant material should be also counted. In this context, it is interesting to note that the gramine presence in soil caused by its leaching from the leaf surface thanks to the rain has been yet described [59]. Generally, both gramine and hordenine have a preferred localization within specific plant organs: gramine has been mainly detected in the first leaf (Table 1) whereas hordenine was highly present in roots (Table 3). This is particularly consistent among all the 20 wild-relative barley accessions studied [43], confirming that wild barley still maintain active both the pathways compared to the modern Barke, in which only the hordenine pathway seems still active. In particular, wild barley accessions from In origin group (Southwestern Iran area) have a high content of these compounds (Tables 1–3), especially for further applications regarding their allelopathic potential in breeding programs, as successfully reported in rice [60]. However, we cannot exclude a translocation of the three metabolites from shoots to roots and vice versa and/or a tissue-dependent biosynthesis without mobilization within the plant. Indeed, it has been observed that sorgoleone is mainly produced and accumulated only in root hairs of *Sorghum* spp [61,62]. A different finding was found for modern spring barley Barke, in which gramine could not be detected in leaves but only in roots (Table 1), while we observed the highest content in roots for hordenine (Table 3). Barke was previously demonstrated as a modern cultivar with a very low gramine content in leaves even though the gene responsible for the last steps of gramine biosynthesis is present and active [36] as also confirmed by the presence of the gramine precursor MAMI by targeted metabolomics analysis (Fig 4). However, from this study it seems that the hordenine pathway has been unintentionally favored against gramine biosynthesis in Barke, as a possible consequence of the domestication or for a "higher" metabolic cost that could have decreased high yields [18]. Gramine and hordenine were already suggested in the last decades as one of the main allelopathic alkaloids in barley responsible for its ability to suppress weeds [18,20,21,63]. Since N-methyltyramine was also detected in the roots exudates in this study, a comparison of the phytotoxic effects of the three compounds involved were also evaluated on lettuce as a target plant, commonly and widely used plant to test allelopathic and phytotoxic effects [64–66]. Interestingly, hordenine at a concentration of 1 mM (equivalent to approx. 10 µmol g$^{-1}$) was much more efficient in reducing the mean root length than gramine at the same concentration (Table 4). However, only gramine was also able to inhibit root elongation at a lower concentration (0.5 mM, equivalent to approx. 5 µmol g$^{-1}$, Table 4). Regarding the hordenine precursor N-methyltyramine, the phytotoxic effects are generally lower compared to hordenine and gramine (Table 4). These results are in contrast with previous studies, in which gramine was

demonstrated to have greater inhibition effects than hordenine [21,67,68]. Yet, these studies observed the toxic effect on different target species. Concentrations used in this phytotoxicity tests were approx. ten times greater than those detected in the root exudates of the barley lines tested (Fig 2). However, in soil conditions allelochemicals could be found more concentrated due to the localized active release in very small rhizosphere soil solutions (a few tens of μL) or localized tissue decomposition. Furthermore, we used buffered alkaloid solutions with MES which might have modified their toxicity [49]. It has in fact been demonstrated that the presence of MES, although having the advantage of more pH stability, could affect roots processes like root exudation [69]. It was also observed that, while the mean root length decreased in the presence of the three allelopathic compounds, the mean root surface and diameter increased, resulting in shorter but thicker roots (Table 4) as previously observed for coumarin on alfalfa [70] and for rye allelochemicals on cucumber seedlings [71]. These results suggest that gramine, hordenine and N-methyltyramine could affect the root development on susceptible plants like lettuce. Negative effects, *i.e.* inhibition of cell division, cell wall destabilization or expansion of the vascular cylinder and cortex cell layer, due to hormone-unbalance or cell wall peroxidases enhancement, or alteration of cell wall composition [72–74], prove their ability as phytotoxic compounds.

## Conclusions

While the mechanisms underpinning the biosynthesis and accumulation of gramine and hordenine in barley tissues remain to be fully elucidated, this study identified the most promising wild barley genotypes to embark in such investigations. For example, our data can guide the selection of families within the NAM to be subjected to further genetic characterization. Likewise, this material and our data can be exploited to formulate and test hypothesis on the wider biological significance of secondary metabolites, such as their implication in modulating the plant microbiota both above- and below-ground.

## Supporting information

**S1 Fig. Unsupervised hierarchical cluster analysis.** Unsupervised hierarchical cluster analysis from the fold-change based heatmaps in leaves and roots of wild relatives and modern cultivated barley.
(PDF)

**S1 Table. Barley accessions by origin group.** Barley accessions by origin group: T-NS = Turkey near Diyarbakir and northern Syria, Iq-In = Northern Iraq and western Iraq, T-G = Turkey near Gaza, L-WS = Lebanon-western Syria, In = Southwestern Iran, Asia = Central Asia, I-J = Israel-Jordan, H.ag = Tibet. GS = Genetic similarity between donor and Barke, based on simple matching.
(PDF)

**S2 Table. Discriminant phenolic compounds identified in leaves.** Discriminant phenolic compounds identified by VIP (Variable Importance in Projection) analysis following OPLS-DA discriminant analysis in Barley leaves. Compounds are provided together with VIP scores (measure of variable's importance in the OPLS-DA model) > 1.6.
(PDF)

**S3 Table. Discriminant phenolic compounds identified in roots.** Discriminant phenolic compounds identified by VIP (Variable Importance in Projection) analysis following OPLS-DA discriminant analysis in Barley roots. Compounds are provided together with VIP

scores (measure of variable's importance in the OPLS-DA model) > 1.6.
(PDF)

## Acknowledgments

We thank Dr Davide Bulgarelli (University of Dundee, UK) for the critical comments on the manuscript.

## Author Contributions

**Formal analysis:** Mauro Maver, Begoña Miras-Moreno, Luigi Lucini, Marco Trevisan, Youry Pii.

**Investigation:** Mauro Maver, Tanja Mimmo.

**Supervision:** Tanja Mimmo.

**Validation:** Luigi Lucini, Youry Pii, Stefano Cesco, Tanja Mimmo.

**Visualization:** Mauro Maver, Begoña Miras-Moreno, Luigi Lucini.

**Writing – original draft:** Mauro Maver.

**Writing – review & editing:** Luigi Lucini, Stefano Cesco, Tanja Mimmo.

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
