## [Decision Letter · Decision Letter 0]

16 Mar 2020

PONE-D-19-34543

New insights in the allelopathic traits of different barley genotypes: Middle Eastern and Tibetan wild-relative accessions vs. cultivated modern barley

PLOS ONE

Dear Mr Maver,

Thank you for submitting your manuscript to PLOS ONE. After careful consideration, we feel that it has merit but does not fully meet PLOS ONE’s publication criteria as it currently stands. Therefore, we invite you to submit a revised version of the manuscript that addresses the points raised during the review process.

We would appreciate receiving your revised manuscript by Apr 30 2020 11:59PM. To enhance the reproducibility of your results, we recommend that if applicable you deposit your laboratory protocols in protocols.io, where a protocol can be assigned its own identifier (DOI) such that it can be cited independently in the future. For instructions see: http://journals.plos.org/plosone/s/submission-guidelines#loc-laboratory-protocols

We look forward to receiving your revised manuscript.

Kind regards,

Zhong-Hua Chen, Ph.D.

Academic Editor

PLOS ONE

Journal Requirements:

2. Please ensure that you include a title page within your main document. You should list all authors and all affiliations as per our author instructions and clearly indicate the corresponding author.

Reviewers' comments:

Reviewer's Responses to Questions

**Comments to the Author**

1. Is the manuscript technically sound, and do the data support the conclusions?

Reviewer #1: Yes

Reviewer #2: Yes

2. Has the statistical analysis been performed appropriately and rigorously? 

Reviewer #1: Yes

Reviewer #2: Yes

3. Have the authors made all data underlying the findings in their manuscript fully available?

Reviewer #1: Yes

Reviewer #2: Yes

4. Is the manuscript presented in an intelligible fashion and written in standard English?

Reviewer #1: Yes

Reviewer #2: Yes

5. Review Comments to the Author

Reviewer #1: The author of this manuscript is trying to explore the three kinds of allelochemicals in barley which could be used for weeding. In this context, the different metabolomic profile in leaves and root were found in 20 accessions；meanwhile these allelochemicals were phytotoxic to the root growth of lettuce. My comments as below:

1. The author should reconstruct the “abstract” logically to make it easy for reading, specifically from line 45-52. And it is better to add one conclusion sentence at the end of the abstract.

2. Line236-238 the figure legend should be before the figure not in the text. The layout should be adjusted to make it good to read.

3. The purpose of this investigate is to use allelochemicals for weeding, so why do you choose lettuce to test the phytotoxic? In current situation, lettuce is not a kind of weed growing in crop field, right?

4. Line 293, in group, should be lower case.

Reviewer #2: In this paper, authors investigated the effects of the gramine and hordenine on the root and leaves in 19 barley cultivars, including one wild barley and one modern spring cultivar barley. The results was interesting and the data could support the findings sufficiently. There are some minor issues in the manuscript: 1. NAM is short for nested associated mapping, authors don’t need to put the full name before the abbreviations; 2. Line 93, “have been observed that…”, who? 3. Line 381 “ in the early 50ies”. Please revise them.

6. PLOS authors have the option to publish the peer review history of their article (what does this mean?). If published, this will include your full peer review and any attached files.

Reviewer #1: Yes: Feifei Wang

Reviewer #2: No

---

## [Author Response · Author response to Decision Letter 0]

20 Mar 2020

As reported in "Response to Reviewers" file uploaded

Dear Editor and referees,

We have appreciated the observations you have raised and we think that, thanks to your suggestions and advice, we could further improve the quality of our work. 

As follows, a point-to-point response to the criticisms raised has been provided. The changings to the text have been added in the “Revised Manuscript with Track Changes” file.

Best regards,

Mauro Maver 

Reviewer # 1:

1. The author should reconstruct the “abstract” logically to make it easy for reading, specifically from line 45-52. And it is better to add one conclusion sentence at the end of the abstract

We amended the abstract, also adding a conclusion sentence, making the abstract easier to read.

2. Line236-238 the figure legend should be before the figure not in the text. The layout should be adjusted to make it good to read.

This has been changed accordingly.

3. The purpose of this investigate is to use allelochemicals for weeding, so why do you choose lettuce to test the phytotoxic? In current situation, lettuce is not a kind of weed growing in crop field, right?

Thanks for the comment. Yes, lettuce is not a weed in crop fields but at the same time, lettuce has been well known and demonstrated as a sensitive plant to phytotoxic substances, and it has been widely used in phytotoxic investigations. Typical plant bioassay species used to investigate ecotoxicity of chemicals and soil amendments to higher terrestrial plants include lettuce: ASTM, 2003; ISO, 2005; OECD/OCDE, 2006a, b

• ASTM (American Society for Testing and Materials), 2003. ASTM E1963 – 02: Standard guide for conducting terrestrial plant toxicity tests. American Society for Testing and Materials, Philadelphia, PA, Philadelphia, PA.

• ISO (International Organization for Standardization), 1993. ISO 11269-1. Soil quality: 828 determination of the pollutants 1 effects on soil flora. Part 1. Method for the measurement of inhibition of root growth. International Organization for Standardization, Geneva. 

• ISO (International Organization for Standardization), 2005. ISO 11269-2. Soil quality: 831 determination of the pollutants effects on soil flora. Part 2. Effects of chemicals on the emergence and growth of higher plants. International Organization for Standardization, Geneva.

4. Line 293, in group, should be lower case.

We added the article “the” to specify the In group and to avoid further misunderstanding.

Reviewer # 2:

1. NAM is short for nested associated mapping, authors don’t need to put the full name before the abbreviations.

This has been changed accordingly.

2. Line 93, “have been observed that…”, who?

This has been changed accordingly..

3. Line 381 “ in the early 50ies”. Please revise them.

This has been changed accordingly.

---

## [Decision Letter · Decision Letter 1]

6 Apr 2020

New insights in the allelopathic traits of different barley genotypes: Middle Eastern and Tibetan wild-relative accessions vs. cultivated modern barley

PONE-D-19-34543R1

Dear Dr. Maver,

We are pleased to inform you that your manuscript has been judged scientifically suitable for publication and will be formally accepted for publication once it complies with all outstanding technical requirements.

With kind regards,

Zhong-Hua Chen, Ph.D.

Academic Editor

PLOS ONE

Additional Editor Comments (optional):

Reviewers' comments:

Reviewer's Responses to Questions

**Comments to the Author**

1. If the authors have adequately addressed your comments raised in a previous round of review and you feel that this manuscript is now acceptable for publication, you may indicate that here to bypass the “Comments to the Author” section, enter your conflict of interest statement in the “Confidential to Editor” section, and submit your "Accept" recommendation.

Reviewer #1: All comments have been addressed

Reviewer #2: All comments have been addressed

2. Is the manuscript technically sound, and do the data support the conclusions?

Reviewer #1: Yes

Reviewer #2: Yes

3. Has the statistical analysis been performed appropriately and rigorously? 

Reviewer #1: Yes

Reviewer #2: Yes

4. Have the authors made all data underlying the findings in their manuscript fully available?

Reviewer #1: Yes

Reviewer #2: Yes

5. Is the manuscript presented in an intelligible fashion and written in standard English?

Reviewer #1: Yes

Reviewer #2: Yes

6. Review Comments to the Author

Reviewer #1: (No Response)

Reviewer #2: In this paper, authors investigated the effects of the gramine and hordenine on the root and leaves in 19 barley cultivars, including one wild barley and one modern spring cultivar barley. The results was interesting and the data could support the findings sufficiently. This paper is ready to publish.

7. PLOS authors have the option to publish the peer review history of their article (what does this mean?). If published, this will include your full peer review and any attached files.

Reviewer #1: Yes: Feifei Wang

Reviewer #2: No

---

## [Editor Report · Acceptance letter]

13 Apr 2020

PONE-D-19-34543R1 

New insights in the allelopathic traits of different barley genotypes: Middle Eastern and Tibetan wild-relative accessions vs. cultivated modern barley 

Dear Dr. Maver:

I am pleased to inform you that your manuscript has been deemed suitable for publication in PLOS ONE. Congratulations! Your manuscript is now with our production department. 

With kind regards,

on behalf of

Dr. Zhong-Hua Chen 

Academic Editor

PLOS ONE